# BridgePolicy: Visuomotor Policy Learning via Stochastic Optimal Control

## Abstract

Imitation learning has been widely used in robotic learning, where policies are derived from expert demonstrations. Recent advances leverage generative models, such as diffusion and flow-based methods, to better capture multi-modal action distributions and temporal dependencies. However, these approaches typically impose conditioning during the forward and reverse process, which inevitably introduces manifold deviation and estimation error. In this work, we propose BridgePolicy, a condition-free generative visuomotor policy that explicitly incorporates observations into the forward process through a diffusion bridge formulation grounded in stochastic optimal control. By sampling actions from observation distributions instead of random noise, BridgePolicy reduces stochasticity and achieves more controllable policy behaviors. However, directly bridging observations to actions poses new challenges, as the action distribution may exhibit mismatched data shape, and the robot observations are inherently multi-modal. In contrast, the diffusion bridge can only connect one-to-one distributions with the same shape. To address the challenges of aligning distributional endpoints and handling multi-modal robot observations, we design a semantic aligner for distribution shape alignment, and a modality fusion module for unifying robot states and visual inputs. Experiments across 52 tasks on 3 benchmarks and 4 real-world tasks demonstrate that BridgePolicy consistently outperforms state-of-the-art generative policies.

## 1 Introduction

Imitation learning (IL) (Osa et al., 2018) is a widely adopted learning paradigm in robotic learning (Li et al., 2024; Shafiullah et al., 2023; Bin Peng et al., 2020), where a robot is provided with a set of expert demonstrations and learns to mimic the provided demonstrations to perform the tasks effectively. Recently, generative models such as diffusion model (Ho et al., 2020; Song et al., 2020) and flow matching (Lipman et al., 2024) gains its prominence in IL owing to their capacity to fit multi-modal distributions and learn sequential correlation (Chi et al., 2023; Ze et al., 2023; Zhang et al., 2025). These methods share a similar principle: they perturb action trajectories into random noise via a forward process defined by a stochastic or ordinary differential equation (SDE or ODE), and then train a neural network conditioned on observations to reverse this process, iteratively transforming noise into executable actions.

Within this learning paradigm, Diffusion Policy (Chi et al., 2023) and 3D Diffusion Policy (Ze et al., 2023), known as DP and DP3, employ an SDE-defined forward process and train the neural network conditioned on visual inputs and robot states to control the denoising process to sample actions from the random noise. FlowPolicy (Zhang et al., 2025), in contrast, employs an ODE-defined forward process, which reduces the stochasticity during training and inference. Despite their successes in learning expert policy, current generative policies heavily rely on conditioning mechanisms (Chi et al., 2023; Ze et al., 2023; Zhang et al., 2025) to steer the sampling process toward observations. This external correction introduces issues such as manifold deviation (Yang et al., 2024) and fitting error (Tang & Xu, 2024).

Diffusion Bridge has demonstrated considerable success in the image generation domain (Yue et al., 2023; Li et al., 2023; Luo et al., 2023; De Bortoli et al., 2021a; Zhou et al., 2023), where it directly modifies the forward process such that the endpoint distribution of the forward process naturally

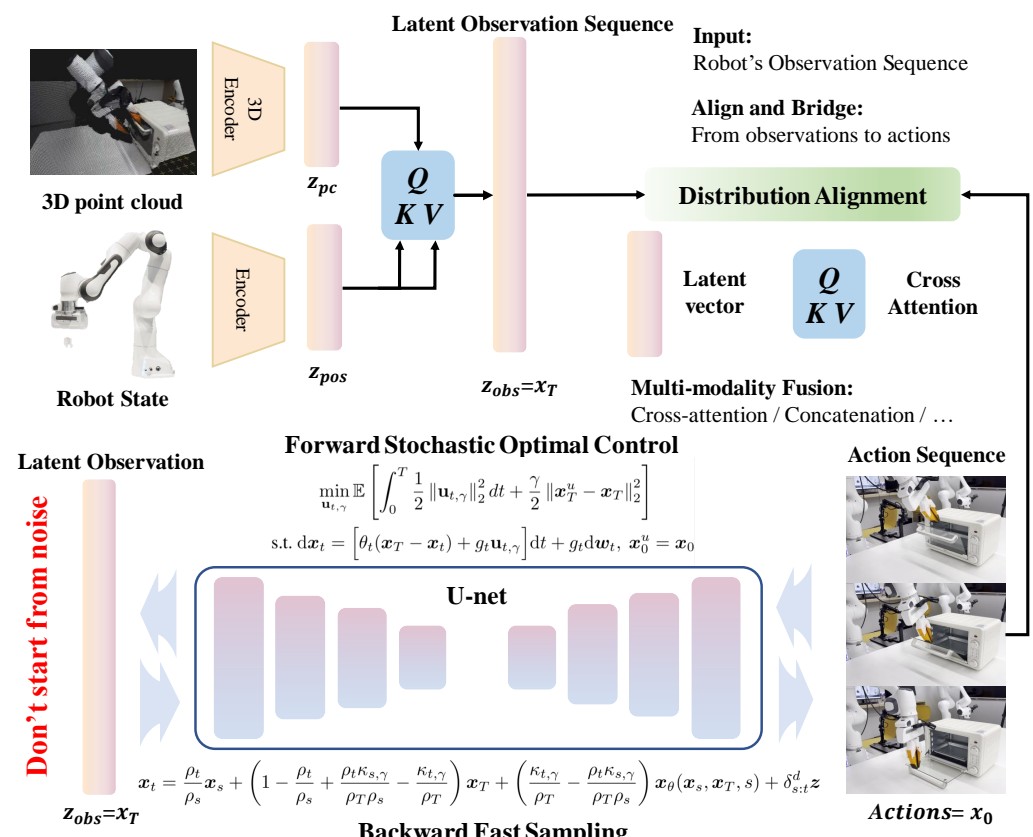

Figure 1: The architecture of BridgePolicy. Its input is the observation consisting of robot state and 3D point cloud. After modality fusion and distribution alignment, BridgePolicy iteratively transforms the observations into actions via the backward SDE.

aligns with the desired conditioned distribution. This modification is accomplished through Doob's $h$-transform, which provides a principled and mathematically exact formulation by modeling the condition into the forward process, thereby circumventing the manifold deviation (Yang et al., 2024) and fitting error that typically arise from condition-based corrections. Building upon these insights, we propose that Diffusion Bridge can also serve as an effective framework for visuomotor policy learning. Specifically, rather than adopting the conventional diffusion model, we formulate policy learning as the problem of learning the diffusion bridge, thus enabling a condition-free generative process. To this end, observations can be explicitly encoded in the forward process via the framework of stochastic optimal control (SOC) (Zhu et al., 2025; Pan et al., 2025), and the corresponding backward process generates actions by sampling directly from the observation distribution rather than from random noise, which reduces the stochasticity, thereby leading to more controllable and reliable policy behaviors.

However, modeling policy learning as the diffusion bridge form raises two key challenges. First, diffusion bridges require the two endpoint distributions to share the same data shape, which is not naturally satisfied between heterogeneous observation and action spaces. Second, robotic observations are inherently multi-modal—encompassing states, visual inputs, and language instructions—while the diffusion bridge is of one-to-one structure that is designed to connect only two distributions.

To address these challenges, we introduce **BridgePolicy**, a new generative policy framework that directly bridges observations to actions instead of starting from noise. Specifically, we design a semantic aligner that transforms the distributions of observations and actions into a common shape while preserving task-relevant semantics. We propose a modality fusion module that unifies multi-modal inputs into a shared representation, enabling diffusion bridging across heterogeneous obser-

vation sources. Finally, we construct a diffusion bridge via the framework of SOC (Zhu et al., 2025) to explicitly model the mapping from observations to actions. Extensive experiments across 52 tasks on 3 benchmarks and 4 real-world tasks demonstrate that BridgePolicy outperforms existing generative policies including DP, DP3, and FlowPolicy, achieving state-of-the-art performance. Our contributions are summarized as follows:

- We propose BridgePolicy, the first condition-free generative policy that directly maps the robot's observations to executing actions, avoiding the manifold deviation issue and fitting error of the conditional diffusion model.
- To address the conflicts between the one-to-one diffusion bridge and the heterogeneous multi-modal robot data, we introduce two key modules: the distribution aligner and the multi-modal fusion module. These modules address the shape mismatch between observation and action spaces and extend the diffusion bridge to multiple heterogeneous modalities, enabling a condition-free diffusion process.
- Extensive experiments across 52 simulation tasks and 3 real-world tasks show that BridgePolicy consistently outperforms prior generative policies and achieves state-of-the-art performance.

## 2 RELATED WORK

### 2.1 GENERATIVE MODELS IN ROBOTICS

Recent years have witnessed the rise of diffusion models and flow matching as state-of-the-art generative paradigms, largely driven by their remarkable success in image generation (Ho et al., 2020; Song et al., 2020; Rombach et al., 2022; Lipman et al., 2022). Due to their capacity of multi-modal expressiveness, they are widely used in reinforcement learning, imitation learning, and motion planning (Janner et al., 2022; Reuss et al., 2023; Chi et al., 2023; Pearce et al., 2023; Sridhar et al., 2023; Xian & Gkanatsios, 2023; Prasad et al., 2024; Saha et al., 2024). In robotics, both diffusion models and flow matching have been utilized as policy frameworks, where action sequences are generated from random Gaussian noise under the condition of 2D or 3D observations via SDE or ODE generative processes, respectively (Chi et al., 2023; Ze et al., 2024; Zhang et al., 2025; Hu et al., 2024). However, our approach is pioneering in its explicit modeling of the direct transition between observation and action distributions via SOC, circumventing the need for conditioning and its associated error sources, thereby enabling robust end-to-end generation of action sequences.

### 2.2 DIFFUSION BRIDGE FOR GENERATIVE MODELING

The diffusion bridge enables the transition between two arbitrary distributions without the need to initiate diffusion from a random Gaussian distribution, which is a more advanced and accurate paradigm for distribution-to-distribution transformation. On the one hand, Diffusion Schrödinger Bridges (Liu et al., 2023; Shi et al., 2024; De Bortoli et al., 2021b; Somnath et al., 2023) aim to determine a stochastic process by solving an entropy-regularized optimal transport problem between two distributions. However, its high computational complexity, particularly pronounced in high-dimensional settings or constraints, poses significant challenges for direct optimization. On the other hand, other works (Zhou et al., 2023; Yue et al., 2023) incorporate Doob's $h$-transform into the forward SDE process, delivering remarkable performance on image restoration benchmarks. UniDB (Zhu et al., 2025) formulates Diffusion Bridge through an SOC-based optimization, proving that Doob's $h$-transform is a special case of SOC theory, thereby unifying and generalizing existing $h$-transform-based diffusion bridges. In this work, we leverage the ability of the diffusion bridges to connect two heterogeneous distributions, specifically the observation distribution and the action distribution of an expert policy, and harness this mechanism for policy learning.

## 3 PRELIMINARIES: DIFFUSION BRIDGE VIA STOCHASTIC OPTIMAL CONTROL

Stochastic Optimal Control (SOC) is a mathematical framework dedicated to deriving optimal control policies for dynamical systems under uncertainty. It has been successfully applied across various domains, including finance (Geering et al., 2010), style transfer (Rout et al., 2024), and

robotics (Rueckert et al., 2014; Elamvazhuthi & Berman, 2015). Denote $\boldsymbol{x}_0$ and $\boldsymbol{x}_T$ as the pre-determined initial state and the terminal respectively, consider the following Linear Quadratic SOC problem:

$$\min_{\mathbf{u}_{t,\gamma}} \mathbb{E}\left[\int_0^T \frac{1}{2}\|\mathbf{u}_{t,\gamma}\|_2^2\, dt + \frac{\gamma}{2}\|\boldsymbol{x}_T^u - \boldsymbol{x}_T\|_2^2\right]$$

$$\text{s.t. } \mathrm{d}\boldsymbol{x}_t = \left[\theta_t(\boldsymbol{x}_T - \boldsymbol{x}_t) + g_t\mathbf{u}_{t,\gamma}\right]\mathrm{d}t + g_t\mathrm{d}\boldsymbol{w}_t,\ \boldsymbol{x}_0^u = \boldsymbol{x}_0, \tag{1}$$

where $\boldsymbol{x}_t^u$ is the controlled state, $\theta_t$ and $g_t$ are some scalar-valued functions with the relation $g_t^2 = 2\lambda^2\theta_t$ and the steady variance level $\lambda^2$ is a given constant, $\boldsymbol{w}_t$ denotes the Wiener process, $\|\mathbf{u}_{t,\gamma}\|_2^2$ is the instantaneous cost, and $\frac{\gamma}{2}\|\boldsymbol{x}_T^u - \boldsymbol{x}_T\|_2^2$ is the terminal cost with its penalty coefficient $\gamma$. UniDB (Zhu et al., 2025) provides the closed-form optimal controller $\mathbf{u}_{t,\gamma}^*$ for the SOC problem (1) as a specific example of its framework as UniDB-GOU (Zhu et al., 2025), whose forward SDE is formed as

$$\mathrm{d}\boldsymbol{x}_t = \left[\theta_t(\boldsymbol{x}_T - \boldsymbol{x}_t) + g_t\mathbf{u}_{t,\gamma}^*\right]\mathrm{d}t + g_t\mathrm{d}\boldsymbol{w}_t,\ \text{with } \mathbf{u}_{t,\gamma}^* = \frac{g_t e^{-2\bar{\theta}_{t:T}}}{\gamma^{-1} + \bar{\sigma}_{t:T}^2}(\boldsymbol{x}_T - \boldsymbol{x}_t), \tag{2}$$

and the transition probability is

$$p(\boldsymbol{x}_t \mid \boldsymbol{x}_0, \boldsymbol{x}_T) = \mathcal{N}(\bar{\boldsymbol{\mu}}_t, \bar{\sigma}_t'^2\mathbf{I}),$$

$$\bar{\boldsymbol{\mu}}_t = \xi_t\boldsymbol{x}_0 + (1 - \xi_t)\boldsymbol{x}_T,\ \xi_t = e^{-\bar{\theta}_t}\frac{\gamma^{-1} + \bar{\sigma}_{t:T}^2}{\gamma^{-1} + \bar{\sigma}_T^2},\ \bar{\sigma}_t'^2 = \frac{\bar{\sigma}_t^2\bar{\sigma}_{t:T}^2}{\bar{\sigma}_T^2}, \tag{3}$$

where $\bar{\theta}_{s:t} = \int_s^t \theta_z dz$, $\bar{\theta}_t = \int_0^t \theta_z dz$, $\bar{\sigma}_{s:t}^2 = \lambda^2(1 - e^{-2\bar{\theta}_{s:t}})$, and $\bar{\sigma}_t^2 = \lambda^2(1 - e^{-2\bar{\theta}_t})$. Meanwhile, its backward reverse SDE is given by

$$\mathrm{d}\boldsymbol{x}_t = \left[\theta_t(\boldsymbol{x}_T - \boldsymbol{x}_t) + g_t\mathbf{u}_{t,\gamma}^* + g_t^2\nabla_{\boldsymbol{x}_t}\log p(\boldsymbol{x}_t \mid \boldsymbol{x}_T)\right]\mathrm{d}t + g_t\mathrm{d}\tilde{\boldsymbol{w}}_t, \tag{4}$$

where the score $\nabla_{\boldsymbol{x}_t}\log p(\boldsymbol{x}_t \mid \boldsymbol{x}_T)$ can be estimated by the noise prediction neural network $-\boldsymbol{\epsilon}_\theta(\boldsymbol{x}_t, \boldsymbol{x}_T, t)/\bar{\sigma}_t'$ and $\tilde{\boldsymbol{w}}_t$ is the reverse-time Wiener process. As for the sampling process, UniDB++ (Pan et al., 2025) provides an acceleration algorithm of (4) and the related updating rule with time steps $0 \leq t < s$ is formed as

$$\boldsymbol{x}_t = \frac{\rho_t}{\rho_s}\boldsymbol{x}_s + \left(1 - \frac{\rho_t}{\rho_s} + \frac{\rho_t\kappa_{s,\gamma}}{\rho_T\rho_s} - \frac{\kappa_{t,\gamma}}{\rho_T}\right)\boldsymbol{x}_T + \left(\frac{\kappa_{t,\gamma}}{\rho_T} - \frac{\rho_t\kappa_{s,\gamma}}{\rho_T\rho_s}\right)\boldsymbol{x}_\theta(\boldsymbol{x}_s, \boldsymbol{x}_T, s) + \delta_{s:t}^d\boldsymbol{\epsilon},$$

$$\delta_{s:t}^d = \lambda\rho_t\sqrt{\frac{1}{e^{2\bar{\theta}_t} - 1} - \frac{1}{e^{2\bar{\theta}_s} - 1}}, \tag{5}$$

where $\boldsymbol{x}_\theta(\boldsymbol{x}_t, \boldsymbol{x}_T, t) = (\boldsymbol{x}_t - (1 - \xi_t)\boldsymbol{x}_T - \bar{\sigma}_t'\boldsymbol{\epsilon}_\theta(\boldsymbol{x}_t, \boldsymbol{x}_T, t))/\xi_t$, $\rho_t = e^{\bar{\theta}_t}(1 - e^{-2\bar{\theta}_t})$, $\kappa_{t,\gamma} = e^{\bar{\theta}_{t:T}}((\gamma\lambda^2)^{-1} + 1 - e^{-2\bar{\theta}_{t:T}})$, and $\boldsymbol{\epsilon} \sim \mathcal{N}(0, I)$. For more details, please refer to (Zhu et al., 2025; Pan et al., 2025).

## 4 METHOD

The BridgePolicy aims to learn a policy $\pi : \boldsymbol{O} \to \boldsymbol{A}$, which predict the actions $\boldsymbol{a} \in \boldsymbol{A}$ given the observations $\boldsymbol{o} \in \boldsymbol{A}$. The observations include 3D point cloud as the visual input and robot state and the actions are chunked into a short sequence of a trajectory finishing the task. The overall framework of BridgePolicy are illustrated in Figure 1. We begin by introducing the formulation of the policy learning as a diffusion bridge problem and then discuss how we address the two key challenges, the multi-modal distribution bridge and the mismatch of the distribution shapes.

### 4.1 DECISION MAKING VIA DIFFUSION BRIDGE

Prior works such as DP and FlowPolicy that formulate the policy as a conditional generative model, where the action is generated from random Gaussian noise under the condition of the observation $\boldsymbol{o}$ (Chi et al., 2023; Ze et al., 2024; Zhang et al., 2025). The observation is treated merely as a conditional signal to guide the neural network during denoising. This sole reliance on the condition

| **Algorithm 1** Training | **Algorithm 2** Inference |
|---|---|
| **repeat** | **Input:** Observations $\boldsymbol{o} = \{\boldsymbol{o}_s, \boldsymbol{o}_{pc}\}$ and pre-trained $\boldsymbol{\epsilon}_\theta$ |
| $\quad$ **Input:** Actions $\boldsymbol{a}$, Observations $\{\boldsymbol{o}_s, \boldsymbol{o}_{pc}\}$, and positive weight factors $\alpha, \beta$. | $\boldsymbol{z}_s, \boldsymbol{z}_{pc} = \mathrm{MLP}_s(\boldsymbol{o}_s), \mathrm{MLP}_{pc}(\boldsymbol{o}_{pc})$ |
| $\quad \boldsymbol{z}_s = \mathrm{MLP}_s(\boldsymbol{o}_s), \boldsymbol{z}_{pc} = \mathrm{MLP}_{pc}(\boldsymbol{o}_{pc})$ | $\boldsymbol{x}_T = \boldsymbol{z}_{obs} = \mathrm{softmax}(\boldsymbol{z}_s \cdot \boldsymbol{z}_s^T / \sqrt{d_s})\boldsymbol{z}_{pc}$ |
| $\quad \boldsymbol{z}_{obs} = \mathrm{softmax}(\boldsymbol{z}_s \cdot \boldsymbol{z}_s^T / \sqrt{d_s})\boldsymbol{z}_{pc}$ | **for** $t = T$ **to** $1$ **do** |
| $\quad \boldsymbol{x}_0 = \boldsymbol{a}, \boldsymbol{x}_T = \boldsymbol{z}_{obs}$ | $\quad \boldsymbol{x}_{t-1} \leftarrow \mathrm{Update}(\boldsymbol{x}_t, \boldsymbol{x}_T, \boldsymbol{\epsilon}_\theta, \boldsymbol{\epsilon}, t)$ from (5) |
| $\quad t \sim \mathrm{Uniform}(\{1, ..., T\})$ | **end for** |
| $\quad$ Compute $\boldsymbol{\mu}_{t-1,\theta}$ and $\boldsymbol{\mu}_{t-1,\gamma}$ from (3) | **Return** Actions $\tilde{\boldsymbol{a}} = \boldsymbol{x}_0$ |
| $\quad$ Compute $\mathcal{L}_{DB}, \mathcal{L}_{align}$ from (6) and (9) | |
| $\quad$ Gradient descent on $\mathcal{L} = \mathcal{L}_{DB} + \alpha\mathcal{L}_{align}$ | |
| **until** Converged | |

injection, however, introduces problems like manifold deviation (Yang et al., 2024) and score-fitting errors, which may degrade action quality and lead to unexpected behaviors.

To address these limitations, we adopt the diffusion bridge, which explicitly model the observation in the generation process, directly connecting the observations to the actions and integrating the observations into the entire generation trajectory. Through such a way, it would eliminate the conditioning efforts, thereby avoiding some potential problems as mentioned above.

Assuming that the action $\boldsymbol{a}$ and the observation $\boldsymbol{o}$ shares the same dimension, we define the endpoints of our forward diffusion bridge process (2) as follows:

- The initial state $\boldsymbol{x}_0$: we set the initial state to be the action, i.e., $\boldsymbol{x}_0 = \boldsymbol{a}$.

- The terminal state $\boldsymbol{x}_T$: we set the terminal state to be the observation, i.e., $\boldsymbol{x}_T = \boldsymbol{o}$, which is a pivotal departure from standard conditional diffusion or flow models, where $\boldsymbol{x}_T$ is typically random Gaussian noise.

Consequently, the goal of our BridgePolicy is to learn a stochastic trajectory that transforms the observation $o$ into a plausible action $a$. To learn such a bridge, we adopt the origin maximum log-likelihood based on the Evidence Lower Bound (ELBO) (Ho et al., 2020) from UniDB (Zhu et al., 2025) as the main training objective. Given a pair of endpoints $(\boldsymbol{x}_0, \boldsymbol{x}_T)$ and the noise prediction model $\boldsymbol{\epsilon}_\theta(\boldsymbol{x}_t, \boldsymbol{x}_T, t)$, the training objective is, specifically,

$$\mathcal{L}_{DB} = \mathbb{E}_{t,\boldsymbol{x}_0,\boldsymbol{x}_T,\boldsymbol{x}_t} \left[ \frac{1}{2g_t^2} \|\boldsymbol{\mu}_{t-1,\theta} - \boldsymbol{\mu}_{t-1,\gamma}\| \right],$$

$$\boldsymbol{\mu}_{t-1,\theta} = \boldsymbol{x}_t - \left( \theta_t + \frac{g_t e^{-2\bar{\theta}_{t:T}}}{\gamma^{-1} + \bar{\sigma}_{t:T}^2} \right)(\boldsymbol{x}_T - \boldsymbol{x}_t) - \frac{g_t^2}{\bar{\sigma}_t'}\boldsymbol{\epsilon}_\theta(\boldsymbol{x}_t, \boldsymbol{x}_T, t), \quad (6)$$

$$\boldsymbol{\mu}_{t-1,\gamma} = \bar{\boldsymbol{\mu}}_{t-1} + e^{-\bar{\theta}_{t-1:t}} \frac{\gamma^{-1} + \bar{\sigma}_{t:T}^2}{\gamma^{-1} + \bar{\sigma}_{t-1:T}^2} \frac{\bar{\sigma}_{t-1}'^2}{\bar{\sigma}_t'^2}(\boldsymbol{x}_t - \bar{\boldsymbol{\mu}}_t).$$

After we obtain the optimal model $\boldsymbol{\epsilon}_\theta^*(\boldsymbol{x}_t, \boldsymbol{x}_T, t)$, the decision making module of our BridgePolicy is implemented as an iterative generation procedure starting from the given observation based on the backward reverse SDE (4) through specially designed accelerating solvers (5) (Pan et al., 2025). The advantage lies in that compared to starting from random Gaussian noise and conditional generation, our sampling process would begin directly from information-rich observations and the observations are directly and globally integrated into the entire trajectory, enabling more controllable generated actions.

While conceptually appealing under a same-dimension assumption, it is still infeasible to directly apply diffusion bridge as the assumption does not exists. We detail how we address the challenges of multi-modal distribution bridging and data shape mismatch.

## 4.2 MODALITY FUSION AND ALIGNMENT

**Encoding the Robot State and Point Cloud.** Our encoding pipeline begins by converting depth images into 3D point clouds, chosen for the efficiency. We then downsample them (512/1024 points in simulation; 2048 in real-world) using farthest point sampling (Qi et al., 2017). Finally, a lightweight MLP with LayerNorm and an MLP state encoder map the point cloud and robot state into a shared latent representation for fusion, which can be formulated as:

$$\boldsymbol{z}_s = \mathbf{MLP}_s(\boldsymbol{o}_s), \quad \boldsymbol{z}_{pc} = \mathbf{MLP}_{pc}(\boldsymbol{o}_{pc}), \tag{7}$$

where $\boldsymbol{o}_s$ is the robot state and the $\boldsymbol{o}_{pc}$ is the point cloud.

**Multi-Modality Fusion.** To enable the diffusion bridge to map between heterogeneous modalities effectively, we introduce the multi-modality fusion module that integrates the 3D point cloud and robot state into a unified representation. This allows the policy to process diverse inputs coherently within our framework. Specifically, we employ a cross-attention mechanism (Rombach et al., 2022) to fuse the modalities, formulated as follows:

$$\boldsymbol{x}_T := \boldsymbol{z}_{obs} = \mathbf{softmax}(\frac{\boldsymbol{z}_s \boldsymbol{z}_s^T}{\sqrt{d_s}})\boldsymbol{z}_{pc}, \tag{8}$$

where $\boldsymbol{z}_s$ is the feature vector of the robot state, $d_s$ is the dimension of $\boldsymbol{z}_s$, $\boldsymbol{z}_{pc}$ is the feature vector of the visual input. Here, $\boldsymbol{x}_T$ is the unified observation feature vector with the same shape of the action chunk, representing the unified observation representation.

**Modality Alignment.** Although multi-modality fusion aligns the observation and action distributions in shape, significant distributional differences still remain and the learned observation representation lacks sampling capability. To address this, we propose using the contrastive learning loss and the KL divergence loss (Kingma & Welling, 2013) to train the aligner. The contrastive loss aligns the semantic proximity of the observation and action distributions, while the KL regularization ensures the observation maintains its sampling ability, facilitating effective policy learning. Specifically, we adopt CLIP loss (Radford et al., 2021) as the contrastive learning loss and use a control factor $\beta$ to control the regularization strength:

$$\mathcal{L}_{align} = \frac{1}{2}(\mathcal{L}_{clip}(\boldsymbol{a}, \boldsymbol{z}_{obs}) + \mathcal{L}_{clip}(\boldsymbol{z}_{obs}, \boldsymbol{a})) + \beta D_{\mathrm{KL}}(\boldsymbol{z}_{obs}, \boldsymbol{\varepsilon}),$$

$$\mathcal{L}_{clip}(\boldsymbol{a}, \boldsymbol{z}) = -\frac{1}{n}\sum_{j=1}^{n}\log\frac{\exp(a_j^\top z_j/\tau)}{\sum_{i=1}^{n}\exp(a_i^\top z_j/\tau)}, \tag{9}$$

where $\boldsymbol{\varepsilon} \sim \mathcal{N}(0, I)$, $n$ is the batch size, $\boldsymbol{z}_{obs}$ is the output of the cross-attention module, $\beta$ is some positive weight, and $\tau$ is the temperature parameter. The entire model involves the diffusion bridge and the modality fusion module and therefore, the overall training objective integrates the diffusion bridge loss (6) with the alignment loss (9):

$$\mathcal{L} = \mathcal{L}_{DB} + \alpha\mathcal{L}_{align}, \tag{10}$$

where $\alpha$ is a positive weight factor. We provide the pseudo-code Algorithm (1) and (2) for the training and inference process of our BridgePolicy. During the training phase, the observations are first fused and reshaped by the modality fusion module and aligner, and then compute alignment loss and diffusion bridge loss with corresponding equations. After training, the observations are fused and reshaped into latent vector, then the latent vector is iteratively updated by the designed accelerating solvers (5) and finally generates the actions.

## 5 EXPERIMENT

### 5.1 SIMULATION EXPERIMENT

**Simulation Benchmark.** We evaluate our BridgePolicy on three benchmarks, Adroit (Rajeswaran et al., 2017), DexArt (Bao et al., 2023), and MetaWorld (Yu et al., 2020). Adroit and DexArt focus on dexterous hand manipulation with varying complexity, while MetaWorld offers a broad spectrum of robotic arm tasks across different difficulty levels.

Table 1: **Main Simulation Results.** Quantitative comparison on success rates among the baselines and our BridgePolicy. We evaluate 50 tasks across 3 benchmarks and report the average success rates of each benchmarks. For MetaWorld, we group the tasks based on their difficulty levels and report the average success rates.

| Methods\Task | Adroit Avg | DexArt Avg | MetaWorld | | | | MetaWorld Avg |
| | | | Easy | Medium | Hard | Very Hard | |
|---|---|---|---|---|---|---|---|
| DP | 0.31±0.17 | 0.45±0.08 | 0.79±0.27 | 0.31±0.24 | 0.10±0.12 | 0.26±0.25 | 0.37 |
| DP3 | 0.68±0.06 | **0.57±0.08** | **0.87±0.20** | 0.61±0.29 | 0.40±0.37 | 0.51±0.33 | 0.60 |
| Simple DP3 | 0.68±0.05 | 0.48±0.07 | 0.86±0.12 | 0.59±0.22 | 0.38±0.26 | 0.47±0.28 | 0.57 |
| FlowPolicy | 0.70±0.12 | 0.54±0.09 | 0.86±0.16 | 0.67±0.21 | 0.59±0.30 | 0.76±0.26 | 0.72 |
| **BridgePolicy** | **0.73±0.10** | 0.56±0.07 | **0.87±0.22** | **0.75±0.28** | **0.63±0.31** | **0.79±0.32** | **0.76** |

**Expert Demonstration Collection.** We collect expert data with script policy in MetaWorld and adopt the VRL3 (Wang et al., 2022) and PPO (Schulman et al., 2017), two reinforcement learning (RL) algorithms, to collect expert data for Adroit and DexArt, respectively. We collect 10 episodes for tasks of the Adroit and MetaWorld benchmarks and 100 episodes of the DexArt benchmark to train the policy.

**Baselines.** For comparison, we select state-of-the-art 2D-based methods DP (Chi et al., 2023) which takes the 2D image as its visual input and 3D conditional diffusion-or flow-based approaches including DP3 (Ze et al., 2024), simple DP3 (its lightweight version), and FlowPolicy (Zhang et al., 2025) which generate the action from Gaussian noise and leverage 3D point cloud as the visual condition in the neural network as baselines.

**Evaluation and Implementation.** We run 3 random seeds for each experiment to avoid possible randomness. For each random seed, we train 3000 epochs and evaluate 20 episodes every 200 training epochs. We pick the five highest success rates per seed and report the mean success rate of the 3 random seeds. For fair comparison, we keep the model architecture consistent with the baselines, the amount of parameters is comparable, and the shared hyperparameters such as the learning rate and weight decay of the optimizer, batch size and training epoch the same as the baselines. Our model is trained on 8 NVIDIA RTX 3090 GPUs. For further details of the implementation of BridgePolicy and other baselines, please refer to Appendix A.2.

**Performance in Simulator.** For Adroit and DexArt, we report the average success rates directly. Particularly for MetaWorld task, it is further categorized into four difficulty levels, with average rates reported for each. The quantitative results are reported in Table 1. Our proposed BridgePolicy demonstrates a consistent performance across all evaluated benchmarks, achieving the highest average success rates. All baseline methods show considerably lower success rates, particularly in more challenging settings (Hard and Very Hard in MetaWorld), which also showcases the robustness of our method in tasks with varying complexity. These results validate the effectiveness of BridgePolicy in leveraging the diffusion bridge formulation for improved policy learning in diverse simulation environments. For success rates of individual tasks, please refer to Appendix A.3.

## 5.2 REAL-WORLD EXPERIMENT

**Experiment Setup and Evaluation.** We evaluate our method on 4 real-world tasks using the Franka Emika Panda robot. The point cloud is acquired using the ZED-2i. The Pick-and-Place task that the gripper pick a bowl and place it in a bucket, the Pouring task that the gripper first grasps the bowl, moves towards an another bowl, pours out the coffee bean in the bowl, and places the bowl on the table steadily, the Oven-Opening task that fully open the oven door and the Oven-Closing task that close the fully opened oven.

We evaluate the policies on 10 episodes of each task. The implementation remains the same as that in the experiments in simulation. We only increase the number of points of the point cloud to 2048 for a more dense representation of the real-world scenario. Please refer to Figure 4 in the Appendix A.4 for the experiment environment.

**Expert Demonstration Collection.** The expert demonstrations are collected by the GELLO human teleopertation system (Wu et al., 2024), manipulated by an experienced graduate. For each task, we collect 50 episodes to train the policy.

Table 2: **Main Real-world Results.** Quantitative comparison of success rates on real-world tasks among the baselines and BridgePolicy.

|  | Oven-Closing | Oven-Opening | Pick Place | Pour | Average |
|---|---|---|---|---|---|
| Simple DP3 | 0.8 | 0.6 | 0.6 | 0.6 | 0.65 |
| DP3 | 0.9 | 0.9 | 0.7 | 0.6 | 0.78 |
| FlowPolicy | **1.0** | 0.7 | 0.5 | 0.1 | 0.58 |
| BridgePolicy | **1.0** | **1.0** | **0.8** | **0.8** | **0.90** |

**Quantitative Comparison on Real-World Task.** The success rates of real-world tasks are shown in Table 2. Under the identical constraint of training with the same dataset of 50 episodes, our BridgePolicy yields the highest success rate on all the tasks, with an average of 0.9, underscoring its effectiveness. Meanwhile, FlowPolicy exhibits a significant performance drop in the real world on tasks requiring location generalization, such as Pouring and Pick-and-Place. DP3 is in the second place with a success rate of 0.78, and Simple DP3 only achieves a success rate of 0.65, possibly due to the small number of parameters.

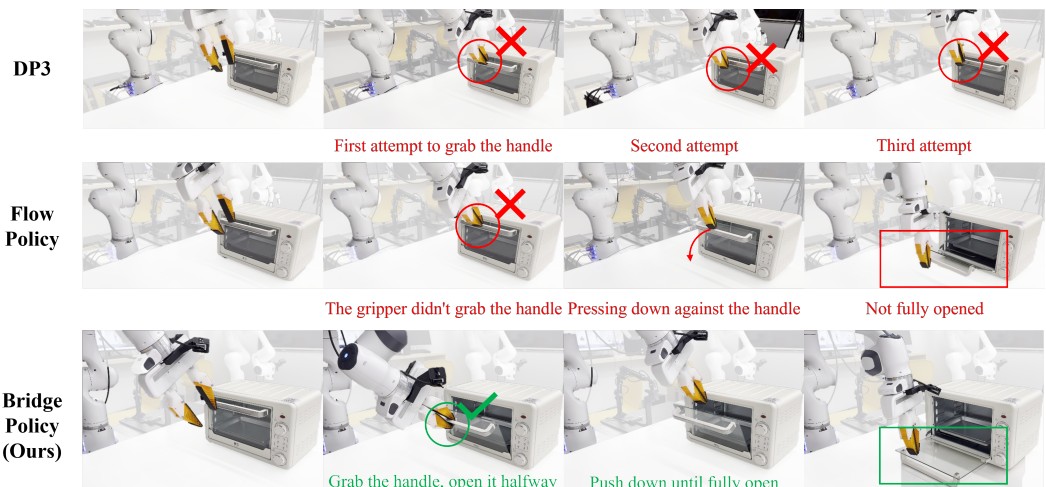

Figure 2: Real-robot comparative visualization of BridgePolicy, FlowPolicy and DP3 at four critical waypoints for the Oven-Opening task.

**Qualitive Comparison on Real-World Task.** As depicted in Figure 2, we select three examples of the real-world task Oven-Opening generated by DP3, FlowPolicy, and BridgePolicy, respectively. From left to right, the sequence presents the key frames of the Franka arm finishing the task. The Oven-Opening task consists of 2 key stages: 1) grabbing the door handle of the oven and opening it halfway; 2) moving the arm upon the half-opened door and pushing it down until it is fully opened. In this very case, DP3 fails to finish the task while FlowPolicy and BridgePolicy both finish the task in the end. BridgePolicy exactly finishes the 2 stages and fully opens the oven, while FlowPolicy fails to half-open the door in the first stage, and it presses down against the door handle directly, leaving the door not fully opened. Videos of the real-world experiments can be found in Supplementary Material.

### 5.3 ABLATION STUDY

**Number of Expert Demonstrations.** The success rate of the agent in accomplishing tasks largely depends on the number of expert demonstrations. Here, we evaluate how the number of expert demonstrations would affect the success rate of the agent finishing the tasks. As shown in Figure 3, we select four tasks for this ablation study: the MetaWorld Pick-place-wall, Box-close, Push, and the Adroit Door task. For all four tasks, increasing the number of expert demonstrations generally improves the agent's success rate. For Pick-place-wall and Push, all policies perform well enough with greater than 30 demonstrations, reaching near-perfect success. BridgePolicy shows the most

Table 3: Quantitative evaluation results on different Adroit tasks with different steady variance levels $\lambda^2$ and penalty coefficients $\gamma$.

| Task | Hyperparameters $(\lambda, \gamma)$ | | | | | |
|---|---|---|---|---|---|---|
| | $(30, 10^7)$ | $(50, 10^7)$ | $(70, 10^7)$ | $(30, 10^5)$ | $(50, 10^5)$ | $(70, 10^5)$ |
| Adroit Hammer | 0.9 | **1.0** | **1.0** | 0.85 | **1.0** | **1.0** |
| Adroit Door | 0.75 | **0.85** | **0.85** | 0.65 | **0.85** | 0.825 |
| Adroit Pen | 0.65 | **0.75** | 0.65 | 0.65 | 0.65 | 0.6 |
| **Avg Success Rate** | 0.77 | **0.87** | 0.83 | 0.72 | 0.83 | 0.81 |

stable and robust performance across all demonstration counts. In Box-close, all methods improve significantly from 10 to 30 episodes, with FlowPolicy and BridgePolicy maintaining high performance at 50 demonstrations.

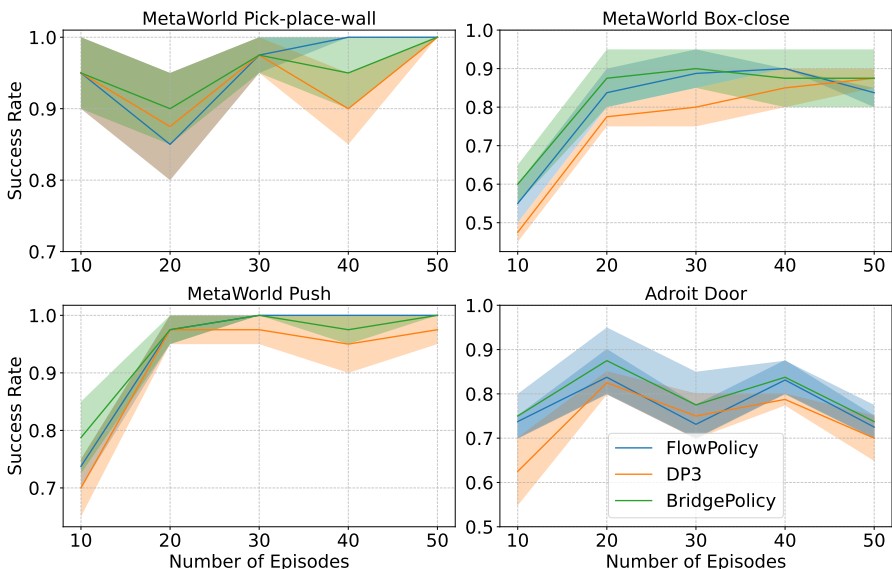

Figure 3: Ablation on the number of demonstrations. We choose four tasks to explore the impact of different numbers demonstrations on BridgePolicy, FlowPolicy and DP3.

**Parameter Sensitivity.** We demonstrate how the key parameters of the BridgePolicy would influence its performance. Specifically, we evaluated different combinations of two parameters: the steady variance level $\lambda^2$ (over 255) and the penalty coefficient $\gamma$ in UniDB (Zhu et al., 2025) with the tasks in the Adroit benchmark. The results are summarized in the Table 3. For Adroit tasks, the combination $(\lambda = 50, \gamma = 10^7)$ emerges as the most robust parameter configuration, achieving the highest average success rate (0.87) and top-tier performance across all individual tasks. Building upon this experience, we use this set of hyperparameters for all the simulation and real-world tasks.

## 6 CONCLUSION

In this work, we introduce BridgePolicy, a generative policy learning paradigm that directly generates action chunks by sampling from the observation distribution. It eliminates the condition efforts of traditional diffusion models, avoiding the manifold deviation and fitting error issues. Extensive experiments on simulation and real-world tasks demonstrate its superiority over the current generative policy. Despite these advancements, the BridgePolicy tends to require more training and inference time than FlowPolicy, partly due to its more complex formulation involving the SDE of a diffusion bridge. In the end, we hope our method helps pave the way towards further research and applications of the diffusion bridge and visuomotor policy learning.

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

# A APPENDIX

## A.1 THE USE OF LLM

At various stages of this study, we use GitHub Copilot to assist in our code implementation and DeepSeek to refine the texts. Besides, Grammarly AI is used to check the syntax errors before our final submission. We confirm that all contents generated by LLM have undergone rigorous review and necessary modifications by the authors to ensure its accuracy and originality.

## A.2 IMPLEMENTATION DETAILS OF BRIDGE POLICY

**Model Architecture** For fair comparison with the baselines, we maximally keep the implementation of the model consistent with the baselines. Specifically, we use the conditional U-Net architecture based on CNN with number of 255M parameters.

For the implementation of the BridgePolicy, our additionally designed aligner and modality fusion modules are significantly lighter than the model itself for predicting the score, with the number of model parameters in this part being less than 1M.

**Training** We split a complete trajectory into segments with a horizon of 4 to create a dataset. The batch size for training is 128. The optimizer is Adam, with a learning rate of $10^{-4}$ and a weight decay of $10^{-2}$. With respect to the schedule of $\theta_t$, we choose a flipped version of cosine noise schedule,

$$\theta_t = 1 - \frac{\cos^2(\frac{t/T+s}{1+s}\frac{\pi}{2})}{\cos^2(\frac{s}{1+s}\frac{\pi}{2})} \tag{11}$$

where $s = 0.008$ is followed from (Yue et al., 2023; Zhu et al., 2025) to achieve a smooth noise schedule.

For BridgePolicy, we train the aligner and modality fusion module jointly with the noise prediction neural network. The Diffusion Bridge hyperparameters we use are $\gamma = 10^7, \lambda = 30$.

## A.3 SUCCESS RATES OF INDIVIDUAL TASKS

| Alg\Task | Adroit | | | DexArt | | | |
|---|---|---|---|---|---|---|---|
| | hammer | door | pen | laptop | faucet | toilet | bucket |
| DP | 0.45±0.05 | 0.35±0.04 | 0.15±0.04 | 0.69±0.08 | 0.23±0.09 | 0.63±0.08 | 0.23±0.08 |
| DP3 | 1.00±0.00 | 0.65±0.05 | 0.40±0.05 | 0.86±0.08 | 0.36±0.04 | 0.73±0.07 | 0.32±0.07 |
| Simple DP3 | 1.00±0.00 | 0.59±0.01 | 0.45±0.03 | 0.79±0.07 | 0.26±0.04 | 0.63±0.07 | 0.22±0.07 |
| Flow | 1.00±0.00 | 0.58±0.06 | 0.53±0.12 | 0.80±0.06 | 0.38±0.09 | 0.70±0.06 | 0.28±0.05 |
| BridgePolicy | 1.00±0.00 | 0.74±0.07 | 0.64±0.10 | 0.83±0.06 | 0.38±0.07 | 0.73±0.07 | 0.28±0.07 |

| Alg\Task | MetaWorld (Easy) | | | | | | |
|---|---|---|---|---|---|---|---|
| | button-press | button-press-topdown | button-press-topdown-wall | button-press-wall | coffee-button | dial-turn | door-close |
| DP | 0.93±0.01 | 0.98±0.01 | 0.96±0.03 | 0.93±0.03 | 0.99±0.01 | 0.63±0.10 | 1.00±0.00 |
| DP3 | 1.00±0.00 | 1.00±0.00 | 0.99±0.02 | 0.99±0.01 | 1.00±0.00 | 0.66±0.01 | 1.00±0.00 |
| Simple DP3 | 1.00±0.00 | 1.00±0.00 | 0.98±0.02 | 0.93±0.02 | 1.00±0.00 | 0.59±0.01 | 0.97±0.00 |
| Flow | 1.00±0.00 | 0.97±0.03 | 0.98±0.02 | 1.00±0.00 | 1.00±0.00 | 0.88±0.10 | 0.90±0.08 |
| BridgePolicy | 0.93±0.06 | 0.92±0.04 | 0.82±0.09 | 1.00±0.00 | 1.00±0.00 | 0.53±0.08 | 1.00±0.00 |

| Alg\Task | MetaWorld (Easy) | | | | | | |
| --- | --- | --- | --- | --- | --- | --- | --- |
| | door-lock | door-open | door-unlock | drawer-close | drawer-open | window-close | window-open |
| DP | 0.75±0.08 | 0.98±0.03 | 0.98±0.03 | 1.00±0.00 | 0.93±0.03 | 0.99±0.01 | 1.00±0.00 |
| DP3 | 0.98±0.02 | 0.86±0.01 | 1.00±0.00 | 1.00±0.00 | 1.00±0.00 | 1.00±0.00 | 1.00±0.00 |
| Simple DP3 | 0.98±0.02 | 0.76±0.08 | 1.00±0.00 | 1.00±0.00 | 1.00±0.00 | 1.00±0.00 | 1.00±0.00 |
| Flow | 1.00±0.00 | 0.66±0.08 | 1.00±0.00 | 0.75±0.01 | 1.00±0.00 | 0.76±0.08 | 0.73±0.05 |
| BridgePolicy | 1.00±0.00 | 1.00±0.00 | 1.00±0.00 | 1.00±0.00 | 1.00±0.00 | 1.00±0.00 | 1.00±0.00 |

| Alg\Task | MetaWorld (Easy) | | | | | | |
| --- | --- | --- | --- | --- | --- | --- | --- |
| | hand-pull | hand-pull-side | lever-pull | peg-unplug-side | reach | reach-wall | plate-slide-back |
| DP | 0.75±0.08 | 0.98±0.03 | 0.98±0.03 | 1.00±0.00 | 0.93±0.03 | 0.99±0.01 | 1.00±0.00 |
| DP3 | 0.98±0.02 | 0.86±0.01 | 1.00±0.00 | 1.00±0.00 | 1.00±0.00 | 1.00±0.00 | 1.00±0.00 |
| Simple DP3 | 0.98±0.02 | 0.76±0.08 | 1.00±0.00 | 1.00±0.00 | 1.00±0.00 | 1.00±0.00 | 1.00±0.00 |
| Flow | 1.00±0.00 | 0.66±0.08 | 1.00±0.00 | 0.75±0.01 | 1.00±0.00 | 0.76±0.08 | 0.73±0.05 |
| BridgePolicy | 1.00±0.00 | 1.00±0.00 | 1.00±0.00 | 1.00±0.00 | 1.00±0.00 | 1.00±0.00 | 1.00±0.00 |

| Alg\Task | MetaWorld (Easy) | | | MetaWorld (Medium) | | | |
| --- | --- | --- | --- | --- | --- | --- | --- |
| | plate-slide-back-side | plate-slide | plate-slide-side | basketball | bin-picking | box-close | hammer |
| DP | 0.99±0.00 | 0.75±0.04 | 1.00±0.00 | 0.85±0.06 | 0.15±0.04 | 0.30±0.05 | 0.15±0.06 |
| DP3 | 1.00±0.00 | 1.00±0.01 | 1.00±0.00 | 0.98±0.02 | 0.38±0.30 | 0.42±0.03 | 0.72±0.04 |
| Simple DP3 | 0.99±0.00 | 1.00±0.01 | 1.00±0.00 | 0.95±0.04 | 0.28±0.16 | 0.38±0.05 | 0.62±0.09 |
| Flow | 0.65±0.05 | 1.00±0.00 | 1.00±0.00 | 0.66±0.06 | 0.66±0.14 | 0.81±0.04 | 0.98±0.02 |
| BridgePolicy | 1.00±0.00 | 1.00±0.00 | 1.00±0.00 | 1.00±0.00 | 0.43±0.06 | 0.61±0.05 | 1.00±0.00 |

| Alg\Task | MetaWorld (Medium) | | | | | | |
| --- | --- | --- | --- | --- | --- | --- | --- |
| | peg-insert-side | push-wall | soccer | coffee-pull | coffee-push | sweep | sweep-into |
| DP | 0.34±0.07 | 0.20±0.03 | 0.14±0.04 | 0.34±0.07 | 0.67±0.04 | 0.18±0.08 | 0.10±0.04 |
| DP3 | 0.67±0.07 | 0.51±0.08 | 0.18±0.03 | 0.85±0.03 | 0.94±0.03 | 0.96±0.03 | 0.17±0.05 |
| Simple DP3 | 0.48±0.07 | 0.38±0.08 | 0.16±0.03 | 0.92±0.12 | 0.86±0.06 | 0.88±0.03 | 0.09±0.05 |
| Flow | 0.70±0.09 | 0.63±0.08 | 0.33±0.05 | 0.96±0.02 | 0.61±0.06 | 0.70±0.04 | 0.31±0.02 |
| BridgePolicy | 0.60±0.08 | 1.00±0.00 | 0.38±0.05 | 0.97±0.01 | 0.93±0.03 | 0.92±0.02 | 0.42±0.02 |

| Alg\Task | MetaWorld (Hard) | | | | |
| --- | --- | --- | --- | --- | --- |
| | pick-out-of-hole | pick-place | assembly | push | hand-insert |
| DP | 0.00±0.00 | 0.02±0.01 | 0.15±0.01 | 0.28±0.03 | 0.09±0.02 |
| DP3 | 0.14±0.09 | 0.12±0.04 | 0.99±0.01 | 0.62±0.03 | 0.14±0.04 |
| Simple DP3 | 0.08±0.06 | 0.12±0.06 | 0.79±0.01 | 0.32±0.03 | 0.12±0.05 |
| Flow | 0.31±0.04 | 0.63±0.05 | 1.00±0.00 | 0.73±0.05 | 0.26±0.02 |
| BridgePolicy | 0.38±0.06 | 0.75±0.02 | 1.00±0.00 | 0.79±0.03 | 0.25±0.06 |

| Alg\Task | MetaWorld (Very Hard) | | | | |
|---|---|---|---|---|---|
| | stick-push | stick-pull | shelf-place | pick-place-wall | disassemble |
| DP | 0.63±0.03 | 0.11±0.02 | 0.11±0.03 | 0.05±0.01 | 0.43±0.07 |
| DP3 | 0.97±0.04 | 0.27±0.08 | 0.19±0.10 | 0.35±0.08 | 0.75±0.04 |
| Simple DP3 | 0.97±0.05 | 0.15±0.08 | 0.05±0.01 | 0.28±0.05 | 0.50±0.03 |
| Flow | 1.00±0.00 | 0.56±0.03 | 0.40±0.03 | 0.95±0.03 | 0.88±0.03 |
| BridgePolicy | 1.00±0.00 | 0.91±0.03 | 0.19±0.03 | 0.96±0.05 | 0.87±0.03 |

## A.4 REAL-WORLD EXPERIMENT SETUP

The real-world experiment setup is shown in Figure 4, the Franka arm is equipped with the FastUMI gripper. Our data collection specification is to collect 15 frames of point cloud, robot state and action per second. To facilitate training, each episode will be padded to a uniform length.

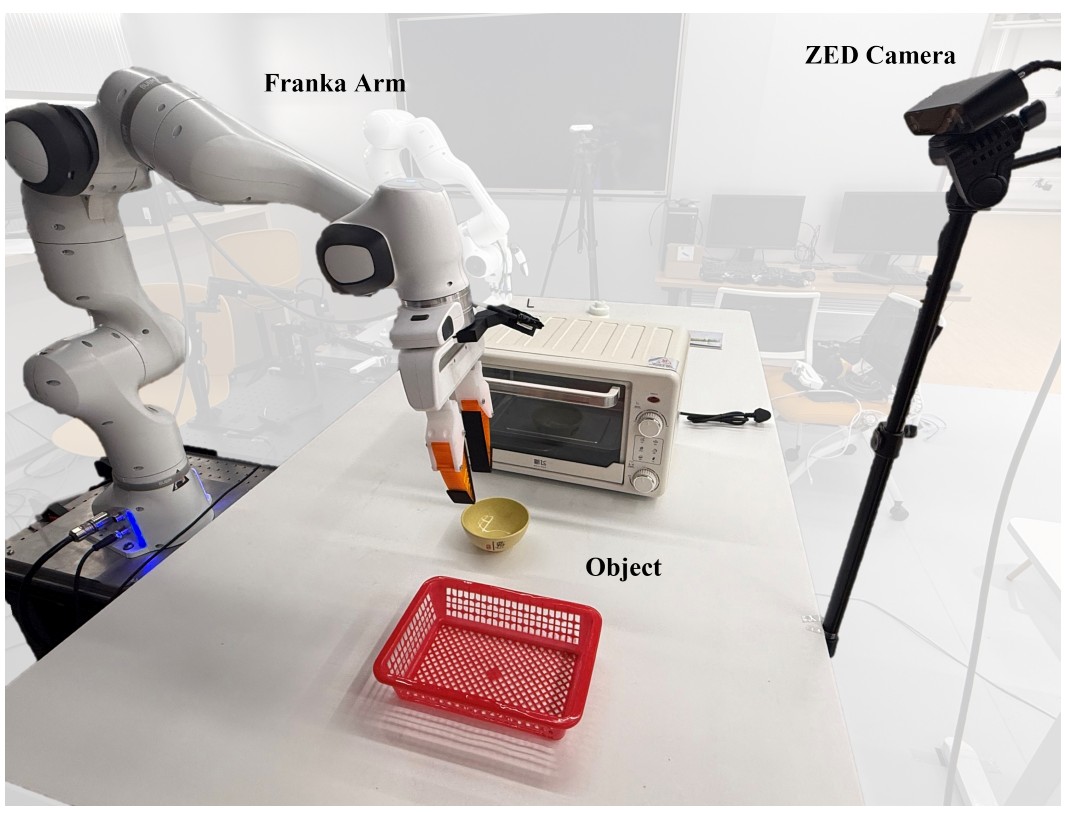

Figure 4: Real-world experiment setup.

