# OpenReview forum: "Bridge Policy: Visuomotor Policy Learning via Stochastic Optimal Control"
_ICLR.cc/2026/Conference — ICLR 2026 Conference Withdrawn Submission_

### Official Review · Reviewer_Ar9m · 2025-10-26

**Soundness:** 3
**Presentation:** 3
**Contribution:** 2
**Rating:** 4
**Confidence:** 3

**Summary:**

The paper introduces a new generative visuomotor policy that is condition-free and maps observations directly to actions through a diffusion bridge grounded in stochastic optimal control. The authors argue that traditional diffusion-based or flow-based imitation learning methods suffer from manifold deviation and the policies learned are less controllable and stable. To address the limitations, the paper reframes the problem and extends diffusion bridge theory from image generation to robot control, which connects observation and action distributions natively, avoiding conditioning. The policy forward starts from observation rather than noise and the backward process generates actions through SOC principles. The experiments show that BridgePolicy improves the success rate by a large margin across different simulation and real-world tasks, demonstrating the effectiveness of the proposed method.

**Strengths:**

- The condition-free idea is reasonable and promising, especially if you want a smoother, deterministic imitation learning policy that can work in a well-observed setting.
- The diffusion bridge formulation is pretty clear, which models the transformation from observation to action using a stochastic differential equation from SOC. It also introduces a semantic distribution aligner that combines contrastive loss and KL regularization to shape mismatch.
- The experimental results are strong enough to show that Bridge Policy improves the task success rate compared to traditional diffusion-based and flow-based methods. With fewer demonstrations, Bridge Policy tends to show better stability and when demonstrations increase, the success also increases.

**Weaknesses:**

- Through the Bridge Policy reduces extrinsic stochasticity and avoids score-fitting instability, it is less flexible in some sense. A condition-free bridge bakes the observation distribution into the diffusion dynamics. Once it is learned, you cannot easily intervene or modulate the policy externally. It is a trade-off between stability and flexibility. For more general policy or complex tasks, some degree of conditioning is crucial.
- The SOC-based formulation leads to higher computational cost, which could constrain long-horizon or high-frequency control.
- In the experiment section, the results demonstrate that Bridge Policy can improve the task success rate, but lacks other metrics to support the improvement of stability, sample efficiency, etc..

**Questions:**

- More ablations or analyses on policy failure modes can help to determine when the bridge misaligns distributions or overfits to certain visual scenes, to better demonstrate the effectiveness of the proposed method. It was not clear in the paper.
- If the diffusion bridge only ever connects distributions seen during training, it might fail catastrophically on unseen combinations of states and visuals, as mentioned in the weaknesses. Can you discuss more?
- It’s unclear how much improvement comes from better representation learning or from the new diffusion formulation. Ablations on the aligner, bridge formulation could be better.
- Typos:

--- Is the Norm missing a square symbol in Eq (6)? It should be squared to match the diffusion ELBO.
--- Should the KL term in Eq (9) be asymmetric? If so, it should be reformulated.

---

### Official Review · Reviewer_ahJj · 2025-10-27

**Soundness:** 2
**Presentation:** 3
**Contribution:** 2
**Rating:** 2
**Confidence:** 4

**Summary:**

The paper introduces BridgePolicy, a new generative model for visuomotor imitation learning. The core idea is to frame policy learning as a diffusion bridge problem, which directly learns a stochastic mapping from the observation distribution to the action distribution. This is in contrast to conventional diffusion policies that generate actions from random noise, conditioned on observations. The authors argue this condition-free approach, grounded in stochastic optimal control, avoids issues like manifold deviation and fitting errors inherent in conditional models. To handle the challenges of bridging heterogeneous, multi-modal distributions, the paper proposes a semantic aligner and a modality fusion module. The aligner uses a combination of contrastive loss and KL regularization to map observations to a latent space with the same shape as the action space. The method is evaluated on 52 simulation tasks across three benchmarks (Adroit, DexArt, MetaWorld) and four real-world tasks, reportedly outperforming state-of-the-art baselines like DP3 and FlowPolicy.

**Strengths:**

1. The central idea of using a diffusion bridge to directly map observations to actions is novel and theoretically elegant. It offers a principled, condition-free alternative to existing generative policies, which could potentially resolve known issues with conditioning schemes.
2. The method achieves state-of-the-art performance across a wide range of simulated tasks. BridgePolicy achieves a 0.90 average success rate compared to 0.78 for the next best baseline in the real-world tasks.
3. Well written, Figure 1 provides a helpful overview of the proposed architecture.

**Weaknesses:**

1. **Central Motivation is Not Directly Validated:** The primary motivation is to avoid "manifold deviation" and "fitting error" from conditioning. However, the paper provides no direct evidence to support this claim. The only evidence is improved task success, which is an indirect and inconclusive metric for this specific claim.

2. **Poor Justification for Key Design Choices in the Aligner:** The proposed alignment loss (Eq. 9) is a core contribution, yet its components are poorly justified. The KL regularization term, $D_{KL}(z_{obs}, \epsilon)$, forces the encoded observation latent distribution towards a standard normal distribution $\mathcal{N}(0, I)$. The paper claims this "ensures the observation maintains its sampling ability," a vague and unsubstantiated statement. This regularization is highly counter-intuitive; it actively penalizes the model for learning the true, complex, multi-modal structure of the observation space, potentially leading to mode collapse. This directly contradicts the goal of leveraging "information-rich observations" and seems more like a VAE-esque regularization hack than a principled choice for a diffusion bridge endpoint. A strong justification for why one would want to destroy the natural structure of the observation manifold is absent.

3. **Insufficient and Unconvincing Experimental Validation:** The empirical evidence, while showing positive trends, is not rigorous enough to support the strong claims of superiority. The simulation results are averaged over only 3 random seeds.

4. **Crucial Ablations are Missing:** The paper fails to ablate the most important new hyperparameters introduced. The alignment loss introduces weights $\alpha$ and $\beta$. The paper's performance is likely highly sensitive to these parameters, yet no ablation study is provided to analyze their impact or justify their chosen values. The parameter sensitivity analysis in Table 3 only investigates $\lambda$ and $\gamma$ from the base UniDB framework, which is insufficient. This is a critical omission that questions the robustness and generality of the method.

**Questions:**

1. Can the authors provide a rigorous theoretical or empirical justification for the KL regularization term $D_{KL}(z_{obs}, \epsilon)$ in Equation 9? Forcing the information-rich observation endpoint $x_T = z_{obs}$ of the diffusion bridge to be distributed as a standard Gaussian seems fundamentally at odds with the goal of the model. Why should the learned latent observation distribution match unstructured noise? Have you investigated the impact of removing this term (i.e., setting $\beta=0$)?

2. The simulation results in Table 1 are based on only 3 seeds and exhibit very high variance, making the performance differences between BridgePolicy and the top baseline (FlowPolicy) statistically questionable. Could the authors please run experiments with more seeds (e.g., 5 or 10) and provide statistical significance tests to properly validate the claim of superior performance?

3. The alignment loss $L_{align}$ introduces two crucial hyperparameters, $\alpha$ and $\beta$. Their values could heavily influence the model's performance. Why was an ablation study on these hyperparameters omitted? Can you provide this analysis and explain how their values were determined for the final experiments?

4. The paper's motivation rests on avoiding "manifold deviation" and "fitting error." Beyond the final task success rates, can you provide any direct evidence that BridgePolicy actually mitigates these specific issues?

5. For the contrastive loss component of $L_{align}$, are negative samples drawn only from within the current batch? If so, have you analyzed the potential impact of false negatives and considered more sophisticated negative sampling strategies to ensure a more robust alignment is learned?

---

### Official Review · Reviewer_VcK5 · 2025-11-01

**Soundness:** 3
**Presentation:** 3
**Contribution:** 3
**Rating:** 4
**Confidence:** 4

**Summary:**

The paper proposes a visuomotor policy that uses a diffusion-bridge formulation with a stochastic optimal control view to connect observations (point clouds, proprioception) and action trajectories. The method replaces standard condition-guided diffusion with a bridge process and adds a multimodal aligner and cross-attention fusion. The authors report results on several simulation suites and a small set of real-robot tasks. Overall, the idea is interesting and the robotics application is well motivated.

**Strengths:**

1.The formulation is interesting and innovative. Casting policy learning as a diffusion bridge with an SOC perspective is a fresh angle and is clearly explained.

2.The paper presents a clean module stack (semantic/shape alignment + multimodal fusion) and provides training/inference details. Additionally, the paper includes some ablation analyses (e.g., demo count sensitivity and hyperparameter sweeps), which help readers understand stability.

3.The method is tested on many simulated tasks and several real-robot tasks; results are generally competitive compared with diffusion-based policy learning methods.

**Weaknesses:**

1.Limited practical gain over strong baselines. On several benchmarks the improvements over diffusion/flow policies are small or mixed.

2.Evidence gap for the main claim. The statement that conditional diffusion inevitably causes manifold deviation/estimation error is only supported by the experiment success rates. I would like to see more results to support your claim, such as distributional distances to expert trajectories, score estimation error, and trajectory-manifold deviation—beyond just success rates.

3.Generalization and robustness. Beyond the success rates, I would like to see more results on the generalization results, such as object, viewpoint and visual generalization.

4.Throughput/latency. The method appears slower than some diffusion/flow counterparts, but there is no speed-accuracy trade-off analysis. The slower inference time may hurt the performance of robots.

I would raise my score if my concerns is solved.

**Questions:**

1.For DP3 and simple DP3, are the number of points exactly the same as in your method (both in sim and real)?

2.Manifold-deviation claim. Can you run a controlled study with identical encoders/U-Nets: (a) conditional diffusion/flow, and (b) your bridge? Report distributional distances to expert trajectories, score estimation error, and trajectory-manifold deviation—beyond just success rates. I would like to see the direct comparison between a conditional flow/diffusion and your bridge to support your claim when controlling most of the network architecture to be similar.

3.Generalization. Do your real-robot tasks include unseen objects/poses/scenes or different cameras/lighting?

---

### Official Review · Reviewer_Hmwo · 2025-11-02

**Soundness:** 3
**Presentation:** 3
**Contribution:** 3
**Rating:** 4
**Confidence:** 4

**Summary:**

This paper introduces Bridge Policy, replacing the conditioning in vanilla diffusion policy with diffusion bridge. Briefly, diffusion bridge modifies the forward process such that the terminal distribution is not a random Gaussian, but a point or a distribution related to the conditioning input (observation embedding in this case). And similarly, the reverse process inference also starts from a latent related to the observation rather than random Gaussian. Bridge Policy encodes robot state and point cloud to a latent of the same shape as the output action chunk, learns a policy, and infers with a fast solver. Sim and real environment experiments show that Bridge Policy outperforms prior state of the art in various settings.

**Strengths:**

- The idea of using diffusion bridge, which has shown promise in visual domains, for policy learning, is well-motivated. The policy architecture design makes sense.
- Strong suite of sim and real environments show that the proposed policy works well in various setups.
- Good set of baselines show that the proposed policy outperforms prior state of the art.

**Weaknesses:**

- Lack of ablations. There are some ablations studying the number of demos and hyperparameter sensitivity, but it would be nice to see an ablation on the alignment loss (no contrastive learning, no KL, original method), and observation encoder (cross-attn vs. simple concat).
- It's unclear from the paper how much of the gain is from the observation module vs. the policy head. If the core claim is that diffusion bridge improves over prior policy heads, then it would be important to keep the observation input treatment constant and compare various policy heads.

**Questions:**

- The main concern that I have is whether the baselines receive the same observation input as BridgePolicy. How are the observations handled in the baseline cases?
- It would be good to see additional ablation experiments: alignment: no contrastive loss, no KL; observation encoder: cross-attention / simple concat.

---

### Note · Authors · 2025-11-12

I have read and agree with the venue's withdrawal policy on behalf of myself and my co-authors.